# Do Large Language Models Respect Contracts? Evaluating and Enforcing Contract-Adherence in Code Generation

## Abstract

Prevailing code generation benchmarks, such as HumanEval+ and MBPP+, primarily evaluate large language models (LLMs) with *pass@k* on functional correctness using well-formed inputs. However, they ignore a crucial aspect of real-world software: adherence to *contracts*—the preconditions and validity constraints that dictate how ill-formed inputs must be rejected. This critical oversight means that existing benchmarks fail to measure, and models consequently fail to generate, truly robust and reliable code snippets. We introduce **PACT**, a program assessment and contract-adherence evaluation framework, to bridge this gap. PACT is the first framework designed to systematically evaluate and enhance contract-adherence in LLM-generated code snippets alongside functional correctness. PACT's contributions are threefold: First, it provides a comprehensive test-suite corpus focused on contract violations, extending HumanEval+ and MBPP+. Second, it enables a systematic analysis of code generation under varied prompting conditions. This analysis demonstrates that augmenting prompts with contract-violating test cases significantly enhance a model's ability to respect contracts compared to using contract description alone. Finally, it introduces novel metrics to rigorously quantify contract adherence in both test generation and code generation. By revealing critical errors that conventional benchmarks overlook, PACT provides the rigorous and interpretable metrics to evaluate the robustness of LLM-generated code snippets in both functionality and contract-adherence.

## 1 Introduction

Test case generation is particularly crucial since it validates correctness, reveals corner cases, and supports automated evaluation. Traditional approaches such as random testing (Cha et al., 2015), symbolic execution (King, 1976; Yoon & Cha, 2024; Cadar et al., 2008), and search-based software testing (Harman & McMinn, 2010; Formica et al., 2024; Burnim & Sen, 2008) have laid the groundwork. Then, various large language model (LLM)-driven methods have recently been investigated, including zero-shot prompting (Chen et al., 2021a; Jaremko et al., 2025), coverage-guided test generation (Ryan et al., 2024; Lemieux et al., 2023; Sapozhnikov et al., 2024), mutation-informed generation (Dakhel et al., 2024; Cajica et al., 2021), slicing-based decomposition (Wang et al., 2024), and satisfiability modulo theories (SMT) solver-based test generation (Peleska et al., 2011; Cadar et al., 2008).

Yet current test case generation approaches (Srivastava & Payer, 2021; Yoon & Cha, 2024) and the resulting benchmarks (Chen et al., 2021a) still judge the quality of both test cases and programs exclusively through pass@k (Chen et al., 2021a). In practice, real-world software is governed by *contracts* (Meyer, 1992)—the preconditions and input validation rules that restrict valid inputs and define expected behavior for ill-formed ones. As a black-box metric, pass@k measures correctness only via input-output correspondence on *well-formed* inputs, which are test cases that have already been filtered to comply with these contracts. Consequently, evaluating code snippets solely by well-formed input-output correspondence without considering contracts leads to an inaccurate assessment, as it overlooks whether the generated code snippets enforce pre-conditions and input validation checks explicitly stated or implicitly included in the specification (Liu et al., 2023).

Typically, most coding competitions such as International Collegiate Programming Contest [1] (ICPC) or International Olympiad in Informatics [2] provide specifications that state the constraints to consider when implementing the function for the task. This description includes not only the functional goal—such as computing a sum or sorting a list—but also a set of explicit contracts, such as input bounds, type restrictions, and required error-handling behaviors. These contracts often phrased in natural language or implied via examples, act as contracts that describe the valid input space and define the expected behavior under edge or invalid conditions. Figure 1 provides a clear illustration, showing how a simple functional task description contains `hidden` contracts that are often overlooked. For instance, a function designed to find duplicate values in a list of numbers, `has_dup(nums)`, has an implicit contract that the input must be a list containing only integers. A correct implementation should therefore include specific assertions—such as `assert isinstance(nums, list)` and `assert all(isinstance(x, int) for x in nums)`—to verify the input's type before proceeding with the functional logic. If such checks are omitted, the program may appear correct on well-formed inputs like `[1, 2, 3]`. However, it will fail to reject ill-formed inputs like `'a'` or `['2', 2]`, thus failing to reflect the intended specification. Ignoring these contracts in code generation poses serious reliability and safety issues. For instance, a module might silently accept an invalid input, such as a negative number where a positive one is expected. By doing so, it can propagate a hidden fault by passing a nonsensical result to another part of the system. This chain reaction often leads to logical errors that are extremely difficult to trace back to their original source.

Therefore, contracts are not optional safeguards but integral parts of the specification that define the boundary of valid behavior. Without them, even code snippets that appear functionally correct cannot be trusted to operate safely and reliably. Empirical studies reveal that LLM-generated code snippets frequently overlook these contracts and often fails to guard against ill-formed inputs that violate them. As a result, a generated code may incorrectly pass tests that it should fail, creating the illusion of correctness under default test cases. This shows that many seemingly correct solutions in fact ignore contractual requirements, underscoring that contract adherence is a fundamental necessity for trustworthy code generation.

We introduce **PACT**, a program assessment and contract-adherence evaluation framework, to bridge this gap. The main focus of PACT is twofold: first, to construct high-quality contract-violating test cases, and second, to use them for a systematic analysis of contract-aware code generation. For test case generation, our approach leverages an SMT solver to systematically explore diverse combinations of contract violations, moving beyond simplistic, single-violation tests.

For example, for a function with the contracts `assert isinstance(r, float)`, `assert r > 0`, and `assert h > 0`, our method can generate an input such as `r = -2.5` and `h = 5`. This test case is guaranteed to satisfy the type contract for `r` and the positivity contract for `h`, while precisely violating only the positivity contract for `r`. By pruning logically inconsistent combinations in advance, the solver ensures that only feasible and semantically valid test cases are generated. For our analysis of code generation, we then use

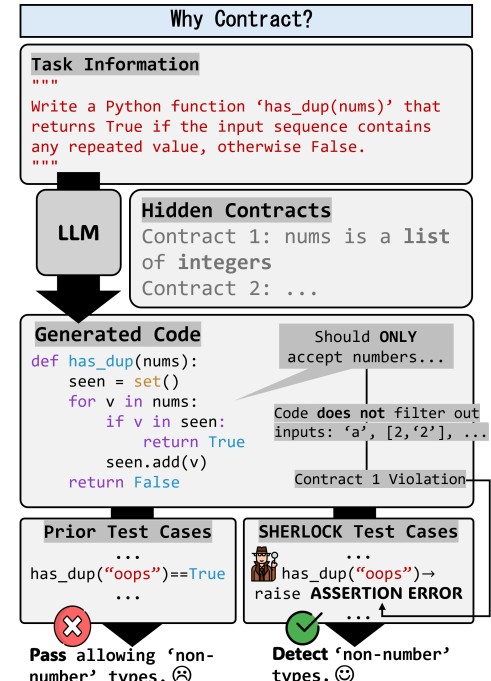

Figure 1: PACT's contract-violating test uncovers an implicit constraint that conventional functional tests miss, proving the need for contract-aware evaluation.

---

[1] ICPC website: https://icpc.global/
[2] IOI website: https://ioinformatics.org/

these precisely targeted test cases to evaluate how different prompting strategies guide an LLM to produce implementations that properly enforce the intended contracts.

In summary, current benchmarks overlook contract violations, thereby inflating the perception of correctness. Addressing this limitation, we propose the PACT framework. PACT extends HumanEval+ and MBPP+ to generate test cases focused on contract violations and introduces novel metrics to quantify contract adherence.

## 2 RELATED WORK

### 2.1 AUTOMATED TEST AND CODE GENERATION

Automated test and code generation have deep roots in traditional software engineering. Early test case generation was typically categorized into black-box methods like random and mutational fuzzing (Godefroid et al., 2005; Claessen & Hughes, 2000), white-box techniques such as symbolic execution (He et al., 2021; Cha et al., 2022; Godefroid et al., 2008), and grey-box approaches like coverage-guided fuzzing (Choi et al., 2019; Stephens et al., 2016; She et al., 2024; Qian et al., 2022). Similarly, early code generation relied on methods like probabilistic grammar-based frameworks (Bielik et al., 2016) and specialized language models (Feng et al., 2020). The advent of the Transformer architecture marked a paradigm shift, establishing LLMs as the dominant approach in both fields (Chen et al., 2021b). The development of modern Code LLMs now typically involves pre-training on code corpora, followed by instruction tuning (Ouyang et al., 2022) and refinement through more advanced techniques like reinforcement learning with execution feedback (Gehring et al., 2025). However, despite these advancements, the primary goal has remained the enhancement of functional correctness, typically measured by pass@k on benchmarks with well-formed inputs.

### 2.2 CONTRACTS

The Design by Contract (DbC) paradigm (Meyer, 1992) argues that software reliability depends on explicitly stated preconditions, postconditions, and invariants, with run-time assertions as the enforcement mechanism. However, mainstream automated testing and existing test-generation benchmarks (Wang et al., 2025; Jain et al., 2025) often overlook latent defects from missing contract checks. While related work on failure-inducing test cases (Zhang et al., 2024; Zhong et al., 2025) is effective at causing general exceptions, this is not the same as verifying adherence to the precise semantics of a given contract. For instance, for a function that requires a list of positive numbers, a generic failure-inducing test might use `None` to cause a `TypeError`, but this does not verify the specific rule that all numbers must be positive. In contrast, a contract-aware test like `[10, 20, -5]` is designed to be caught precisely by an assertion checking for positive values. This distinction highlights the need for systematic methods that can precisely target formal contract specifications rather than just triggering arbitrary errors.

### 2.3 SMT SOLVER

SMT (Satisfiability Modulo Theories) solvers are powerful engines for determining the satisfiability of complex logical formulas across various theories, such as arithmetic and strings (Barrett & Tinelli, 2018). They typically interface using SMT-LIB, a standardized formal language, with Z3 being a widely adopted implementation in automated testing (de Moura & Bjørner, 2008). Traditionally, techniques like symbolic execution have employed SMT solvers to find inputs that satisfy path constraints, focusing on functional coverage over well-formed inputs. In this paradigm, constraints are used as admission checks to filter out invalid data rather than as explicit targets for evaluation. In contrast, our framework leverages SMT solvers to systematically generate ill-formed inputs that precisely violate formalized contracts, allowing us to rigorously test for robustness. A detailed example of how we formulate contracts is provided in Appendix D.

## 3 The Need for Contract-Aware Evaluation

### 3.1 Why Contracts Matter: Blind Spots in Current Benchmarks

Recent studies on code and test case generation (Korraprolu et al., 2025; Sung et al., 2025; Li et al., 2022) have dominantly relied on *pass@k* to evaluate functional correctness. While effective, these metrics, along with approaches that target failure-inducing inputs (Peng et al., 2018), share a common limitation: they primarily operate within the bounds of a program's *legal* input space Dyck et al. (2023). This practice has led to a critical blind spot in popular benchmarks such as HumanEval+ and MBPP+. By design, these benchmarks explicitly filter out and discard any test case that violates a program's pre-conditions (Austin et al., 2021). Consequently, the evaluation process certifies that a program behaves correctly on well-formed inputs, but it fails entirely to assess the program's robustness against ill-formed ones. This results in an incomplete and often inflated assessment of code quality, praising solutions that may be superficially correct but are fundamentally fragile.

This is where the software contracts become essential. As explained in Section 1, contracts are the rules that define the boundary of valid behavior. They are not optional safeguards but a core component of trustworthy software that specifies how a program must identify and reject invalid data. By ignoring contract adherence, existing benchmarks overlook a crucial dimension of software reliability.

### 3.2 Contract-Based Test Paradigm

Merely enlarging a single pool of valid inputs cannot reveal whether a model understands the boundary of the specification it needs to satisfy. The main focus, therefore, lies in constructing contract-violating test cases, which systematically explore the extent to which models enforce contract rules. A contract-violating test case is an input that violates one or more predicates from a set of contracts while satisfying the remaining specification. The reference implementation is augmented with runtime assertions for every predicate in the contract set, and a candidate program passes such a test only when it raises an error consistent with this augmented reference. Introducing contract-violating inputs uncovers false negatives that purely functional tests overlook and provides a rigorous measure of whether a program properly enforces contractual rules.

### 3.3 Task Setup

Each benchmark task consists of a natural language description, a set of contracts, and a functional implementation. We use HumanEval+ and MBPP+, where contract predicates are stroed as assertion literals outside the prompt and reference code. Our task concerns contract-violating test generation and contract-aware code generation. For test generation, we automatically construct a compact set of contract-violating test cases that target specific contract rules and remain feasible with respect to the remaining rules for each benchmark task. These tests are used for evaluating whether LLM-generated code appropriately follows contracts. For contract-aware code generation, we generate code under two prompt conditions, where the first condition is a contract specification (CS) prompt, which includes the functional description and a natural language paraphrase of the contracts. The second condition is an example-augmented specification (EAS) prompt, which is augmented with the contract-violating test cases.

## 4 Methodology

We propose `PACT`, a program assessment and contract-adherence evaluation framework designed to systematically evaluate and enhance the ability of LLMs to generate contract-compliant code. This framework consists of two main stages. First, we generate *contract-violating test cases* to rigorously assess whether LLM-generated code snippets enforces both functional specifications and explicit contracts. Second is the systematic analysis of code generation, where we use these test cases under different prompting conditions to evaluate a model's contract awareness in detail. Unlike prior approaches that rely solely on functionality-based evaluation, PACT extends the evaluation paradigm with contract-violation tests, enabling a more precise and reliable analysis of contract adherence.

Figure 2: Running example of PACT for code generation.

## 4.1 Contract-Violating Test-Case Generation

Direct LLM-based generation is inadequate for contract-aware test case construction. It must consider all subsets of the contract set when producing violating inputs. If a task has $n$ contracts, the number of non empty violation combinations is $2^n - 1$. Direct prompting often misses required combinations and yields contradictory inputs that violate unintended contracts and fails to ensure feasibility under the specification. LLMs also lack a built in mechanism to verify that exactly the targeted contracts are violated while all others hold. We design an SMT-based approach to make these checks accurate and efficient. Our generation procedure is a two-step pipeline for contract-aware test case generation. First, an inference model translates natural language contracts into an Algebraic Data Type (ADT) program. This ADT format is the critical component: it provides a rigid, formal schema for the complex and often nested constraints found in contracts. This ensures that the subsequently generated rules for an SMT solver are syntactically valid and semantically precise. Second, given the ADT, the SMT solver constructs contract violation test cases (CVTs) by breaking one or more specified contracts while ensuring that all remaining contracts are satisfied. The solver validates satisfiability for each candidate combination and extracts concrete models— specific assignments of values that satisfy the constraints— to instantiate inputs for every valid case. We provide a running example of this procedure in Appendix D.

## 4.2 Contract Metrics for Test Case

Unlike standard test cases for functional correctness, test cases for contracts should be the negative samples, violating the contracts and triggering the corresponding assertions. As the concept of contract-violating test cases is different from the standard, we design metrics to analyze whether the generated test cases appropriately correspond to input contracts.

Let $A = \{a_1, \ldots, a_n\}$ be the set of all contract assertions and $T = \{t_1, \ldots, t_m\}$ be the generated test cases. $F_t \subseteq A$ is the set of violated assertions when executing a test case $t \in T$. Finally, $T_{neg} \subseteq T$ represents the set of test cases that successfully violated at least one contract.

**Assert Violation Coverage (AVC)** AVC quantifies the coverage of assert statements that are successfully violated by all the test cases.

$$AVC = \frac{|\{a_i \mid \exists t \in T_{neg} : a_i \in F_t\}|}{n}.$$

The value of 1.0 for AVC ensures that all contract assertions are captured by the test cases. Lower values expose the unexplored regions of the contracts from the test cases.

**Target Specificity (TS)** TS evaluates the precision of each test case by measuring how accurately it violates its intended set of target contracts. To formalize this, we first define $V \subseteq A$ as the set of contracts that a given test case $t \in T_{neg}$ is intended to violate. We then measure the alignment between this intended set $V_t$ and the actually violated set $F_t$ using the Jaccard Index. The final TS

score is the average of these individual precision scores across all negative test cases:

$$TS = \frac{1}{|T_{neg}|} \sum_{t \in T_{neg}} \frac{|F_t \cap V_t|}{|F_t \cup V_t|}.$$

A score of 1.0 indicates perfect precision, where every test case violates exactly the set of contracts it was designed to. Lower scores reveal a discrepancy, indicating that test cases either failed to trigger their intended violations or caused unintended, collateral violations.

### 4.3 CONTRACT-AWARE CODE GENERATION

This section details our methodology for systematically evaluating and enhancing the ability of LLMs to generate code snippets that robustly enforces contracts. To achieve a fine-grained understanding of a model's contract awareness, we investigate its performance under two distinct prompting strategies. Frist, we establish a baseline condition, **Contract Specification (CS)**. For this strategy, we use a powerful Commercial LLM to naturally integrate the contract rules from the HumanEval+ and MBPP+ datasets into the main text of the prompt. This creates a comprehensive prompt that describes both the functional goal and its contractual constraints in natural language. Second, we introduce an enhanced condition, which we term **Example-Augmented Specification (EAS)**. This strategy builds upon the CS prompt by augmenting it with a single, precisely targeted contract- violating test cases for each described contract. This provides a concrete example of what constitutes a violation, intended to guide the model toward more robust code generation.

To measure the impact of these prompting strategies, we assess the generated code snippets using a comprehensive suite of metrics for both functional correctness and contract adherence. Functional correctness is measured using the standard pass@k metric over a set of valid test cases. Contract adherence is evaluated from two perspectives: runtime enforcement using test cases and static alignment of the generated code snippets with the ground-truth contracts, as detailed in Section 4.4.

### 4.4 CONTRACT METRIC FOR CODE GENERATION

For the evaluation of the generated code snippets, we employ pass@k to measure functional correctness on valid inputs and use AVC to measure the correctness of the generated contract assertions. We also design two additional metrics to more precisely evaluate contract-aware code generation.

Let $A = \{a_1, \ldots, a_n\}$ be the set of ground-truth contract assertions, and let $\hat{A} = \{\hat{a}_1, \ldots, \hat{a}_m\}$ be the set of assertions extracted from the LLM-generated code. Let $M \subseteq A \times \hat{A}$ be the set of pairs $(a_i, \hat{a}_j)$ where the ground-truth contract $a_i$ and the generated assertion $\hat{a}_j$ are determined to be semantically equivalent.

**Assertion Alignment Recall (AAR)**  AAR measures the model's ability to implement all required contracts without omission. It is the proportion of ground-truth contracts that are successfully covered by at least one assertion in the generated code, functioning as a recall metric.

$$AAR = \frac{|\{a_i \in A \mid \exists \hat{a}_j \in \hat{A} : (a_i, \hat{a}_j) \in M\}|}{n}.$$

A high AAR score indicates that an output code ensures that all required contract specifications are generated.

**Assertion Alignment Precision (AAP)**  AAP measures the accuracy of the generated assertions, penalizing irrelevant or hallucinated ones. It is the proportion of generated assertions that correspond to a valid ground-truth contract, functioning as a precision metric.

$$AAP = \frac{|\{\hat{a}_j \in \hat{A} \mid \exists a_i \in A : (a_i, \hat{a}_j) \in M\}|}{m}.$$

A high AAP score indicates that the code does not contain assertions of unnecessary or incorrect checks.

## 5 EXPERIMENTAL SETTINGS

### 5.1 DATASETS

Our study utilizes HUMANEVAL+ and MBPP+ benchmarks (Liu et al., 2023), which are enriched with assertion-level contracts. However, these benchmarks have a critical limitation for evaluating code robustness: the contract specifications are neither included in the model prompts, nor are contract-violating inputs included in the official test suites. This evaluation setup exclusively tests for functional correctness on well-formed inputs, leading to an inflated perception of code quality. We construct a supplementary dataset containing contract-violating test cases, as described in Section 4.1.

### 5.2 CANDIDATE MODELS

We utilize o4-mini as our primary test case generator. For our code generation experiments, we evaluate a set of open-source models including gemma-3-12B-it (gemma-3), Deepseek-R1-Distill-Qwen-14B (DeepSeek-R1), Qwen3-14B (Qwen-3), and Phi-4-reasoning-plus (Phi-4-plus).

## 6 EMPIRICAL STUDIES

Our empirical study is structured over three research questions (RQs) designed to evaluate the PACT framework from multiple perspectives. The evaluation is conducted on HumanEval+ and MBPP+. We begin by assessing the quality and precision of the CVTs generated by PACT (RQ1). Next, we investigate whether providing these concrete test cases is more effective for eliciting contract-awareness than using abstract natural language descriptions alone (RQ2). Finally, we analyze the resulting trade-off between the enforced contract adherence and the functional correctness of the generated code snippets (RQ3).

Table 1: Evaluation results of the test case generation on HumanEval+ and MBPP+.

| Benchmark | Method | AVC (↑) | TS (↑) | AVG (↑) |
|---|---|---|---|---|
| HumanEval+ | o4-mini | **97.14**% | 75.60% | 86.37% |
| | o4-mini + SMT Solver | 95.53% | **85.81**% | **90.67**% |
| MBPP+ | o4-mini | **94.67**% | 69.11% | 81.89% |
| | o4-mini + SMT Solver | 93.50% | **84.54**% | **89.02**% |

### 6.1 RQ1: HOW EFFECTIVE IS PACT IN GENERATING HIGH-QUALITY CONTRACT-VIOLATING TEST CASES?

Our first research question investigates the effectiveness of our proposed framework, PACT, in generating high-quality CVTs. A naive baseline approach, which we refer to as o4-mini in Table 1, uses an LLM o4-mini for direct test case generation from given contracts. This method is often inadequate, as it disregards dependencies among contracts and produces logically inconsistent violations, making it difficult to precisely verify individual contract predicates.

In contrast, PACT (o4-mini+ SMT Solver) employs a more robust two-stage approach. First an LLM converts contracts into rules for an SMT solver. Subsequently, an SMT solver uses these rules to generate test cases. This process guarantees the generation of logically sound test cases that are precisely targeted to violate a specific subset of contracts while adhering to the rest.

The empirical results in Table 1 validate the superiority of PACT. While both methods achieve high AVC scores, PACT significantly outperforms the direct generation in TS, achieving over 10%p increases on both HumanEval+ and MBPP+. This demonstrates that our two-stage approach successfully give test cases precisely for intended contracts. The baseline's slightly higher AVC is an expected outcome, as direct generation often triggers a wide range of assertion errors indiscriminately, artificially inflating coverage. **PACT**, on the other hand, focuses on valid, targeted violations.

While PACT generates valid rules, providing a precise test for the input contracts, there are still errors. A case-specific analysis of minor errors that occur even with well-formed rules is detailed in Appendix A.

Table 2: Evaluation results of contract adherence in code generation on HumanEval+.

| Model | Mode | pass@1 (↑) | AVC (↑) | AAR (↑) | AAP (↑) | AVG (↑) |
|---|---|---|---|---|---|---|
| gemma-3 | CS | **84.41**% | 24.85% | 11.41% | 14.04% | 32.79% |
| | EAS | 82.94% | **91.02**% | **28.07**% | **27.77**% | **57.45**% |
| DeepSeek | CS | **73.78**% | 44.12% | 15.65% | 16.97% | 37.63% |
| | EAS | 71.77% | **79.29**% | **27.62**% | **28.01**% | **51.67**% |
| Qwen3 | CS | **78.92**% | 28.04% | 13.17% | 22.55% | 35.67% |
| | EAS | 77.83% | **87.81**% | **31.53**% | **36.09**% | **58.31**% |
| Phi-4-plus | CS | **72.23**% | 52.91% | 18.78% | **21.09**% | 41.25% |
| | EAS | 67.08% | **69.50**% | **21.33**% | 20.06% | **44.49**% |

Table 3: Evaluation results of contract adherence in code generation on MBPP+.

| Model | Mode | pass@1 (↑) | AVC (↑) | AAR (↑) | AAP (↑) | AVG (↑) |
|---|---|---|---|---|---|---|
| gemma-3 | CS | 78.56% | 57.99% | 17.50% | 17.93% | 41.49% |
| | EAS | **78.60**% | **95.57**% | **32.29**% | **31.82**% | **59.57**% |
| DeepSeek | CS | **62.53**% | 64.20% | 17.59% | 17.57% | 40.47% |
| | EAS | 60.15% | **86.70**% | **28.23**% | **27.94**% | **45.47**% |
| Qwen3 | CS | 72.41% | 70.86% | 21.09% | 22.99% | 46.84% |
| | EAS | **72.63**% | **94.85**% | **31.54**% | **32.30**% | **57.83**% |
| Phi-4-plus | CS | **64.89**% | 67.33% | 24.26% | 24.65% | 45.28% |
| | EAS | 63.76% | **74.88**% | **29.20**% | **28.95**% | **49.20**% |

## 6.2 RQ2: Are Concrete Test Cases More Effective than Abstract Descriptions for Eliciting Contract-Awareness?

We compared two prompting methods across various models in Tables 2 and 3. CS, our baseline, provides only an abstract natural language description of the contracts. In contrast, EAS augments the CVTs, generated from Section 4.1, to the CS prompt.

The results demonstrate that for all of our base models, EAS is a more powerful method for eliciting contract-aware code generation. EAS achieves 33.7%p, 11.3%p, and 9.4%p improvements for AVC, AAR, and AAP, respectively, over CS. For models such as Qwen3, DeepSeek-R1, and Phi-4-plus, switching from CS to EAS leads to a dramatic improvement across all contract adherence metrics. This indicates that concrete examples of CVTs provide a clear and unambiguous signal that forces the model to move beyond purely functional logic. The increase in AAR and AAP indicates that the models are correctly generating test cases required for assertion checks, while the relatively huge improvements in AVC show that the resulting codes are more robust against a wider range of invalid inputs.

Concrete examples of contract violations provide a clear, unambiguous signal that forces the model to move beyond purely functional logic and implement the explicit enforcement of contracts—the predefined agreements on how ill-formed inputs must be rejected. For concrete examples of the code snippets and their corresponding LLM-generated test cases, please refer to Appendix B. However, this heightened focus on contract enforcement introduces a notable trade-off, as the model may generate more complex code that impacts its performance on purely functional correctness metrics. This trade-off is analyzed in further detail in the following section RQ3.

## 6.3 RQ3: How Does Enforcing Contract Adherence Impact the Functional Correctness of Generated Code?

While the RQ2 established that augmenting prompts with CVTs significantly enhances a model's contract adherence, this section investigates the resulting trade-off with respect to its functional

correctness. Our analysis reveals that a heightened focus on contract enforcement often comes at the cost of a measurable decline in performance on standard functional correctness benchmarks. This trade-off can be attributed to a shift in the model's optimization focus. When provided with only a functional description, the LLM's sole objective is to produce code snippets that passes functionality-focused tests. However, when the prompt is augmented with CVTs, the model must simultaneously satisfy two competing objectives: generating functionally correct logic and enforcing the specified contractual preconditions. This dual objective increases the complexity of the code generation task. The model is compelled to allocate part of its reasoning capacity to interpreting and implementing the contract rules, which can lead to subtle errors or oversights in the core functional logic. A concrete example of this is presented in Appendix C. Consequently, while the resulting code snippets is more robust and secure against invalid inputs, it may exhibit a lower pass@1 rate on test suites composed entirely of valid, well-formed inputs.

In conclusion, our findings reveal an inherent tension between contract adherence and functional correctness in LLM-based code generation. The pursuit of robustness through contract enforcement comes at a tangible cost to functionality. This trade-off highlights the critical need for advanced training paradigms—such as the reinforcement learning with multi-objective rewards proposed by PACT—capable of optimizing for both objectives simultaneously.

## 7    CONCLUSION

We introduce PACT, the first framework to redefine code and test case correctness by evaluating adherence to task specifications through both functionality and contract-based behavior. While prior benchmarks assess only pass@k on well-formed inputs, PACT introduces a comprehensive paradigm with dual test suites—one for functionality and one for contract violations—along with specific metrics to analyze contract awareness and uncover latent defects, enabling a more principled evaluation of code robustness. Our empirical evaluation demonstrates the effectiveness of PACT across multiple dimensions. We first show that PACT's SMT-solver-based test case generation method ensures more accurate CVTs than direct generation with over 10%p better performance. Furthermore, our results reveal that augmenting prompts with these CVTs is a highly effective strategy for generating robust and contract-aware code, achieving 18.13%p increase in contract-specific metrics and 12.8%p in total average. Our empirical studies on various LLMs demonstrate the effectiveness of PACT, achieving approximately , but also uncovers a critical trade-off between this enhanced contract adherence and a model's performance on functional correctness. These findings confirm that PACT provides a more complete and realistic assessment of an LLM's contract-aware code generation capabilities, moving beyond the limitations of existing benchmarks.

## 8    FUTURE WORK DIRECTIONS

While this work focuses on evaluating contract adherence, a natural next step is to actively improve it through advanced training methodologies. A promising direction is to leverage the novel metrics introduced in our PACT framework as direct training signals. The novel metrics introduced in our PACT framework, such as the runtime metric Assert Violation Coverage (AVC) and the static metrics Assertion Alignment Recall (AAR) and Precision (AAP), are particularly well-suited for this purpose and could be integrated into a multi-objective reward function for RL. This approach could enable models to learn to navigate the trade-off between functional correctness and contract adherence more effectively, optimizing for both objectives simultaneously.

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

## A    CASE STUDY: LOGICAL CONTRADICTIONS IN DIRECT LLM TEST CASE GENERATION

**HumanEval**    In Figure 3 shown in the code snippet, this task includes three sequential contracts: `assert_0` checks if the input is a list, `assert_1` verifies that all elements in the list are strings, and `assert_2` ensures that all strings consist only of digits. A critical dependency exists between these contracts. Specifically, `assert_2` can only be evaluated if `assert_1` is satisfied, because the `isdigit()` method is only valid for string types. A test case designed to violate `assert_1` while satisfying `assert_0` and `assert_2` would therefore be a logically contradictory combination, as a non-string element would cause a `TypeError` before `assert_2` could be checked. Despite this, a direct LLM generation approach often produces such invalid combinations. For instance, when tasked to generate test cases, the LLM produces inputs such as `[123, "456"]`,`["789", [0]]`, and `["456", false]`. These examples fail to isolate a specific contract violation. This highlights a fundamental weakness of the approach, as the LLM tends to generate simplistic contract-violation test cases that fail to respect the logical relationships among contracts.

**Mbpp**    In Figure 4 shown in the code snippet, this task includes four main contracts, which can be grouped by their dependency. The initial contracts, `assert_0` and `assert_1`, perform type checking to verify that both inputs are of a numeric type, such as an integer or a floating-point number. The subsequent contracts, `assert_2` and `assert_3`, check numeric properties, such as ensuring the numbers are positive or fall within a specific range. A critical dependency exists between these

groups of contracts. Specifically, the numeric property checks in `assert_2` and `assert_3` can only be evaluated if the type checks in `assert_0` and `assert_1` are satisfied. For example, a non-numeric type like a `string` or `null` cannot be evaluated for properties like being positive. Therefore, creating a contract-violation test case that violates the initial type contracts (`assert_0` or `assert_1`) while simultaneously satisfying the subsequent property contracts (`assert_2` and `assert_3`) is a logical impossibility. Despite this, a direct LLM generation approach often produces such logically flawed combinations. For instance, when tasked to generate test cases, the LLM produces inputs such as `["abc", null]`, `[null, "abc"]`, `[[1], {"x":1}]`, and `[{"r":1}, [2]]`. Crucially, while these examples successfully violate the initial type contracts, they all inherently fail to satisfy `assert_2` and `assert_3`, yet they are generated as if such a combination were possible. This highlights a fundamental weakness of the approach, as the LLM tends to generate simplistic contract-violation test cases that fail to respect the logical relationships among contracts.

---

**HumanEval/113**

```python
def odd_count(lst):
    assert type(lst) == list, "invalid inputs" # $_CONTRACT_$
    assert all(isinstance(s, str) for s in lst), "invalid inputs" # $_
        CONTRACT_$
    assert all(s.isdigit() for s in lst), "invalid inputs" # $_
        CONTRACT_$

    ans, template = [], "the number of odd elements in the string i of
         the input."
    for s in lst:
        odd_cnt = len(list(filter(lambda ch: int(ch) % 2 == 1, s)))
        ans.append(template.replace("i", str(odd_cnt)))
    return ans

"""
Contract List:
assert_0: assert type(lst) == list, "invalid inputs
assert_1: assert all(isinstance(s, str) for s in lst), "invalid inputs
assert_2: assert all(s.isdigit() for s in lst), "invalid inputs
"""
```

Figure 3: Code and contracts for HumanEval.

## B  CASE STUDY: CONTRACT ENFORCEMENT UNDER DIFFERENT PROMPTING CONDITIONS

We present a direct comparison of code snippets that the DeepSeek model generated for the **MBPP/11** task under two prompting conditions. This comparison illustrates why providing CVTCs is more effective than relying on natural language descriptions alone.

Figure 5 shows the code snippets produced when the model received only the natural language prompt. The prompt contains enough information to infer all necessary contracts. It specifies two explicit contracts, namely that the first input must be a non empty string and that the second input must be a string of length one.It also implies two type contracts that both inputs must be strings. The generated code snippet correctly implements the explicit length based contracts with `assert len(s) > 0` and `assert len(char) == 1`. However, it omits the implicit type checks and treats the word "string" as descriptive context rather than a strict precondition. As a result, the function fails to enforce the contracts under non string inputs.

In contrast, Figure 6 displays the output when the prompt was augmented with concrete CVTCs such as `remove_Occ(2.0, 3.0)`. These test cases provide an unambiguous and executable signal that non string inputs are invalid and must be rejected. The model generates a more robust implemen-

**MBPP/731**

```
import math
def lateralsurface_cone(r,h):

    assert isinstance(r, (int, float)), "invalid inputs" # $_CONTRACT_$
    assert isinstance(h, (int, float)), "invalid inputs" # $_CONTRACT_$
    assert r > 0, "invalid inputs" # $_CONTRACT_$
    assert h > 0, "invalid inputs" # $_CONTRACT_$
    l = math.sqrt(r * r + h * h)
    return math.pi * r  * l

"""
Contract List:
assert_0: assert isinstance(r, (int, float)), "invalid inputs
assert_1: assert isinstance(h, (int, float)), "invalid inputs
assert_2: assert r > 0, "invalid inputs
assert_3: assert h > 0, "invalid inputs
"""
```

Figure 4: Code and contracts for MBPP.

tation that now includes the crucial `isinstance(s, str)` and `isinstance(char, str)` assertions.

This comparison shows that natural language descriptions may contain all required information, yet their inherent ambiguity can prevent the model from adhering to the contract. CVTCs make these implicit contracts explicit and enforceable. They compel the model to produce code snippet that is more comprehensive and adheres to the intended contracts.

**MBPP/11**

```
"""
Write a python function that removes the first and last occurrence of
    a given character from the string; the first input must be a non-
    empty string and the second input must be a string of length one.
"""

def remove_Occ(s, char):
    assert len(s) > 0, "First input must be a non-empty string."
    assert len(char) == 1, "Second input must be a string of length
        one."

    first = s.find(char)
    last = s.rfind(char)

    if first == -1:
        return s

    return s[:first] + s[first+1:last] + s[last+1:]
```

Figure 5: Code generated by DeepSeek with the contract specification (CS) prompt.

```
MBPP/11

"""
Write a python function that removes the first and last occurrence of
    a given character from the string; the first input must be a non-
    empty string and the second input must be a string of length one.

# Contract Test Cases:
>>> remove_Occ(2.0, 3.0)
"AssertionError: invalid input"
>>> remove_Occ(5.0, 4.0)
"AssertionError: invalid input"
>>> remove_Occ('', 2.0)
"AssertionError: invalid input"
>>> remove_Occ(2.0, '')
"AssertionError: invalid input"
"""

def remove_Occ(s, char):
    assert isinstance(s, str) and len(s) > 0, "invalid input"
    assert isinstance(char, str) and len(char) == 1, "invalid input"

    first_occ = s.find(char)
    last_occ = s.rfind(char)

    if first_occ == -1:
        return s

    return s[:first_occ] + s[first_occ+1:last_occ] + s[last_occ+1:]
```

Figure 6: Code generated by DeepSeek with the example-augmented specification (EAS) prompt.

## C    CASE STUDY: CONTRACT ADHERENCE VS. FUNCTIONAL CORRECTNESS

A case study from the **HumanEval/142** task provides a concrete example of the trade-off between contract adherence and functional correctness. The figures 7–9 compare the ground-truth implementation (golden label), the code generated from a contract natural language prompt (CS), and the code generated from a prompt augmented with contract-violating test cases (CVTs).

Figure 8 displays the code generated from the NL-only prompt, which describes both functional requirements and contractual constraints. It fails to translate the natural language contract descriptions into the necessary assertions, it successfully generates code snippets that passes all standard functionality tests. This outcome suggests the model prioritized achieving functional correctness—even developing a logic for handling multiples of 12—at the complete expense of implementing the required robustness checks, leaving the code snippets vulnerable to invalid inputs.

In contrast, Figure 9 shows the code generated when the prompt was augmented with CVTCs. Here, the model successfully implements the necessary assertions, demonstrating strong contract adherence. However, this heightened focus on contract enforcement directly compromises its functional correctness. The model generates a flawed conditional logic that explicitly excludes common multiples of 3 and 4. For an index like 0 or 12, the code incorrectly defaults to the else block, failing to square the number as required by the specification in the Golden Label (Figure 7).

This example clearly illustrates the trade-off. When compelled to handle the dual objectives of implementing contracts and functional logic, the model's reasoning capacity is divided. The added complexity of enforcing contracts leads to a critical oversight in the core functional logic, resulting in code snippet that is robust but functionally incorrect. This underscores the inherent tension between these two competing objectives in LLM-based code generation.

```
HumanEval/142

def sum_squares(lst):
    assert type(lst) == list, "invalid inputs" # $_CONTRACT_$
    assert all(type(x) == int for x in lst), "invalid inputs" # $_
        CONTRACT_$

    ans = 0
    for i, num in enumerate(lst):
        if i % 3 == 0:
            ans += num ** 2
        elif i % 4 == 0:
            ans += num ** 3
        else:
            ans += num
    return ans
```

Figure 7: The ground-truth implementation, correct in both functionality and contracts.

```
HumanEval/142

def sum_squares(lst):
    total = 0
    for i in range(len(lst)):
        num = lst[i]
        if i % 3 == 0 and i % 4 == 0:
            total += num ** 2
        elif i % 3 == 0:
            total += num ** 2
        elif i % 4 == 0:
            total += num ** 3
        else:
            total += num
    return total
```

Figure 8: A functionally correct implementation that fails to enforce contracts.

## D   FORMALIZING CONTRACTS INTO SMT-LIB: TEMPLATE AND A CASE STUDY

This section details the structure of the SMT-LIB template used by our PACT framework. SMT-LIB is a standardized, text-based language used to interface with SMT solvers. It provides a formal syntax for declaring variables, defining functions, and asserting logical formulas, allowing complex problems to be translated into a format that a solver can systematically analyze for satisfiability. Our framework leverages this language to translate nuanced, natural language contracts into a formal representation that can be reasoned about with logical precision.

Figure 10 shows the base template we designed for this purpose. It is composed of several key components, each serving a distinct role in the test generation process. The placeholders within this template are populated as follows:

- **CANONICAL PYTHON-LIKE ADT:** This fixed block defines a universal data structure for representing common Python types. This allows the SMT solver to reason about various input types in a standardized way.

**HumanEval/142**

```python
def sum_squares(lst):
    assert isinstance(lst, list), "AssertionError: invalid input"
    for elem in lst:
        assert isinstance(elem, int), "AssertionError: invalid input"

    total = 0
    for index, num in enumerate(lst):
        if index % 3 == 0 and index % 4 != 0:
            total += num ** 2
        elif index % 4 == 0 and index % 3 != 0:
            total += num ** 3
        else:
            total += num
    return total
```

Figure 9: A robust implementation that enforces contracts but fails on functionality.

- **HELPER FUNCTIONS:** This section is populated with custom functions needed to define the contracts for a specific task. For example, a function to check if a string contains only digits would be defined here.
- **INPUT:** The input variables for the function under test are declared here.
- **BASIC_STRUCTURE:** This section defines fundamental structural constraints on the inputs, such as ensuring a variable is a list composed of integer values.
- **CONTRACT_DEFS:** The specific logical rules of each contract are translated into formal predicates in this section.
- **COMBINATION:** This is the core logic for generating a test case. It contains assertions stating which contracts must be satisfied and which must be violated. The SMT solver then attempts to find a concrete model that satisfies this exact combination of constraints.

Figure 12 shows the ground-truth Python implementation for the **HumanEval/11** task, which requires a function that takes two binary strings of equal length. The SMT-LIB formalization of these requirements is shown in Figure 11. The three assert statements in the Python code directly correspond to the three formal contracts defined in SMT-LIB:

- **C0** verifies that both inputs are strings, corresponding to the `assertion assert isinstance(a, str) and isinstance(b, str)`.
- **C1** ensures their lengths are equal, corresponding to `assert len(a) == len(b)`.
- **C2** checks that they are valid binary strings using a custom isBinaryString helper function, corresponding to `assert set(a).issubset("0", "1") and set(b).issubset("0", "1")`.

The COMBINATION block determines the goal of the test case generation. By choosing to assert either the contract itself, such as `(assert (C0))`, or its negation, such as `(assert (not C0))`, for each rule, this block can instruct the SMT solver to find a test case for any desired combination of contract satisfactions and violations. The specific instance in the figure 11, for example, asserts the negation of all three contracts to generate a test case that violates every precondition simultaneously.

**The SMT-LIB template**

```
ADT_BASE_TEMPLATE = """
(set-logic ALL)

; ==== CANONICAL PYTHON-LIKE ADT (DO NOT MODIFY) ====
(declare-datatypes ((Value 0)) (
  ((IntVal (ival Int))
   (FloatVal (fval Real))
   (StrVal (sval String))
   (BoolVal (bval Bool))
   (Nil)
   (Cons (head Value) (tail Value)))
))

; === ADD HELPER FUNCTIONS HERE ===
<<HELPER_FUNCTIONS>>

; === Inputs ===
<<INPUT>>

; === BASIC STRUCTURE ===
<<BASIC_STRUCTURE>>

; === Contract predicates ===
<<CONTRACT_DEFS>>

; === COMBINATION ===
<<COMBINATION>>

(check-sat)
(get-model)
"""
```

Figure 10: The SMT-LIB template used for formalizing contracts.

**The SMT-LIB template**

```
ADT_BASE_TEMPLATE = """
(set-logic ALL)

; ==== CANONICAL PYTHON-LIKE ADT (DO NOT MODIFY) ====
(declare-datatypes ((Value 0)) (
  ((IntVal (ival Int))
   (FloatVal (fval Real))
   (StrVal (sval String))
   (BoolVal (bval Bool))
   (Nil)
   (Cons (head Value) (tail Value)))
))

; === ADD HELPER FUNCTIONS HERE ===
(define-fun Safe_Sval ((x Value)) String
  (ite (is-StrVal x) (sval x) ""))
(define-fun isBinaryString ((s Value)) Bool
  (and (is-StrVal s)
       (str.in.re (Safe_Sval s) (re.* (re.union (str.to.re "0") (str.
          to.re "1"))))))

; === Inputs ===
(declare-const a Value)
(declare-const b Value)

; === BASIC STRUCTURE ===

; === Contract predicates ===
(define-fun C0 () Bool (and (is-StrVal a) (is-StrVal b)))
(define-fun C1 () Bool (= (str.len (Safe_Sval a)) (str.len (Safe_Sval
    b))))
(define-fun C2 () Bool (and (isBinaryString a) (isBinaryString b)))

; === COMBINATION ===
(assert (not C0)
(assert (not C1)
(assert (not C2)

(check-sat)
(get-model)
"""
```

Figure 11: An example of the SMT-LIB template populated for HumanEval/11

**HumanEval/11**

```python
from typing import List

def string_xor(a: str, b: str) -> str:

    assert isinstance(a, str) and isinstance(b, str), "invalid inputs"
        # $_CONTRACT_$
    assert len(a) == len(b), "invalid inputs" # $_CONTRACT_$
    assert set(a).issubset({"0", "1"}) and set(b).issubset({"0", "1"})
        , "invalid inputs" # $_CONTRACT_$

    return "".join(str(int(a[i]) ^ int(b[i])) for i in range(len(a)))
```

Figure 12: The ground-truth implementation in HumanEval/11

