# OpenReview forum: "Do Large Language Models Respect Contracts? Evaluating and Enforcing Contract-Adherence in Code Generation"
_ICLR.cc/2026/Conference — ICLR 2026 Conference Withdrawn Submission_

### Official Review · Reviewer_beZY · 2025-10-27

**Soundness:** 2
**Presentation:** 2
**Contribution:** 2
**Rating:** 2
**Confidence:** 4

**Summary:**

The paper constructs a benchmark and studies LLM contract-adherent code generation, i.e., generated code should reject invalid inputs with leading assertion lines.
For benchmark construction, it sets up a pipeline where an LLM converts task contracts into a structured constraint form, and an SMT solver uses those constraints to synthesize contract-violating test cases.
This work evaluated the contract-adherent code generation task on four LLMs.
The results demonstrated that assertion lines with some validation ability can be generated in many cases.

**Strengths:**

- Timely focus on an important problem. The paper targets contract-aware test generation for LLM code, which is increasingly relevant in practical settings.

- Constructive exploration of solver-assisted generation. By contrasting LLM-only vs. LLM+SMT, the work advances a pragmatic "neuro-symbolic" approach—letting learned models propose structure while a solver enforces logical rigor—a broadly useful paradigm that would carry over to other program-analysis and testing settings.

- Attention to robustness gaps. The emphasis on inputs that violate intended contracts (while satisfying others) highlights brittleness in naive tests and current benchmarks.

**Weaknesses:**

- Performance drop. The authors discussed the trade-off between contract quality and functional correctness. However, no quantitative analysis was provided.
This drop from the original code generation to this work's *contract-adherent code generation* would be significant.
For example, [Phi-4-plus](https://huggingface.co/microsoft/Phi-4-reasoning-plus) has 92.3% Pass@1 on HumanEval+, but only 72.23% in this work's CS setting and 67.08% in the EAS setting.
Functional correctness drop is an important concern when proposing the new task named *contract-adherent code generation*.
It would be better if this question could be further analyzed.

- Reliance on incomplete contracts from EvalPlus.
The paper treats EvalPlus contracts as ground truth, but EvalPlus does not ensure contract completeness.
In EvalPlus, these contracts are byproducts used to filter invalid inputs, and not violating the contract does not imply the input is legally admissible.
EvalPlus offers no guarantee on the completeness of contract sets.
However, this work built both generation and evaluation directly on EvalPlus contracts, and does not provide any completeness or soundness checks.
This can inflate or distort results (e.g., an input avoiding contract clauses while still producing semantically invalid), bias metrics like TS/AVC.

- Limited contribution due to reliance on manually crafted contracts.
The approach and its reported gains hinge on the availability of contract sets.
This restricts the method’s portability: the proposed benchmark construction approach cannot transfer to any datasets that lack contracts without first incurring substantial human effort to author them.
Further,  the paper's assertion evaluation relies on the manual-crafted contracts' quality, while evaluating the contract suite's quality would be a valuable open question. Treating existing contract corpora as ground-truths sidesteps this open problem and narrows the work’s broader impact.

- Misaligned metrics (AVC vs. AAP).
The paper’s metric design (AAR/AAP) appears ill-suited to the task.
In Tables 2 and 3, AVC is consistently far higher than AAP.
For example, under gemma-3/EAS the Pass@1 is 82.94% and AVC is 91.02%, indicating that a large share of generated contracts are both sensible and effective at filtering erroneous inputs.
However, AAP is 27.77%.
This gap suggests two issues: (1) the ground-truth contract set is likely incomplete, so many valid generated clauses go unmatched; and (2) the AAR/AAP "matching to ground truth" procedure may not be an appropriate way to assess the utility of a contract set.
The paper should analyze this discrepancy directly (e.g., manual audits of "AAP-missed but AVC-hit" cases, ablations with relaxed/semantic matching, etc.).
Without such analysis, AAP (and any conclusions that rely on it) risks undervaluing useful contracts and obscuring what the method is actually improving.

- Limited contribution due to limited extensibility to richer data and environments.
The SMT solver-assisted pipeline worked on simple data types (HumanEval-style scalars/lists).
However, scaling to more realistic settings, repository-level execution, stateful APIs, complex object graphs, etc., would require substantial additional modeling. Therefore, this approach can only work on simple datasets and does not show potential for extension, limiting its contribution.

- The metric name "AVG" is confusing. It is the shortening of "average". However, the meaning is not explained in the paper, and there is a similar name, "AVC", people would think "AVG" is a related metric to "AVC", but cannot find an explanation about it, which would be very confusing.

**Questions:**

1. What is the average size of the contract set from Evalplus ($|A|$ in your definition)? What is the average size of your generated test cases for each task ($|T|$ in your definition)?
2. A metric that combined Pass@1 and AVC would be more reasonable. Since people check AVC, they can only know the generation result's performance on CVTs, but do not know if it is functionally correct as well. It would be better if the AVC number did not account for functionally incorrect generation results. Could it be designed?

---

### Official Review · Reviewer_eTFa · 2025-10-30

**Soundness:** 3
**Presentation:** 2
**Contribution:** 2
**Rating:** 4
**Confidence:** 5

**Summary:**

This paper introduces PACT, a “program assessment and contract-adherence evaluation framework”. PACT follows two steps: 1) using an LLM, it reads a natural language problem and constructs an Algebraic Data Type (ADT) program (essentially rules for an SMT solver to follow), and 2) an SMT solver (z3) finds counterexamples and these are saved as contract violation (CVT) test cases. Critically, the solver creates the CVT cases by breaking the rules in a fashion that each rule is tracked such that at the end, we may have a CVT for each candidate combination (ex, if there are n rules, then there are 2^n - 1 combinations). The authors use these CVT to study two kinds of prompting techniques: 1) zero-shot (called Contract Specification or CS in the paper), and 2) single-shot (Example-Augmented Specification, EAS). Finally, they propose some metrics for the tests generated (Assert Violation Coverage or AVC and Target Specificity or TS), and for the code generated when using the contracts (Assertion Alignment Recall or AAR and Assertion Alignment Precision or AAP). Finally, they use HumanEval+ and MBPP+ benchmarks to bootstrap the process as both benchmarks have problem statements, code solutions, and contracts specified as assertions.

**Strengths:**

* I think the paper is well motivated. Although the introduction and related work could be organized more clearly, you can see that the authors identify the various works that have gone into test generation for code including areas from fuzzing to LLMs. This helps to establish that current LLM benchmarks do not check for robustness to malformed inputs and that contracts do matter in production software.
* Their idea of SMT-guided synthesis from contract ADTs seems novel and really is a great idea. This design achieves higher TS over using purely LLM generated tests.
* The metrics proposed are natural and the authors perform enough evaluations to validate their design. They also offer a good qualitative analysis of their metrics through their 3 research questions.

**Weaknesses:**

* The paper is a bit disorganized. The introduction goes through 2 examples yet the running example on line 246 which could have helped discussions is kicked out to the appendix. There is a really large focus on motivating the problem which comes at the cost of implementation. We don’t really get to the meat of PACT until the end of page 4, i.e., nearly half of the submission is motivating contracts. Code contracts can cover preconditions, postconditions, and invariants. This work feels preliminary as the authors make no mention of why postconditions and invariants are ignored despite “contracts” appearing more than 10 times on page 1 alone.
* The experiments as is are a little tough to reproduce. The LLM experiments from section 5.2 do not specify decoding settings such as temperature and it is not clear to me from Figure 10 if that is sufficient to produce the ADTs in PACT or if there are still parts missing.
* EAS selects on the CVTs but I do not understand which is selected from the 2^n - 1 choices and why. Even if it’s random or arbitrary, it should be called out at least if not ablations performed on which CVT and the number of CVTs.
* I can tell from the results that EAS boots AVC, AAR, and AAP across models but I am not entirely sure if this is due to genuine contract reasoning or some prompt anchoring. Some controls like random negative examples, or synthetic non-contract negatives might help with this or even some justification.
* There are a few typos in the paper like “stroed” on line 196.

**Questions:**

* In Section 4.4 we are interested in the semantic equivalence between the ground-truth contracts and the generated assertions. How exactly is that checked? Do you have a matcher that can account for something like communitivity (ex, assert a ^ b == assert b ^ a) ?
* What is the reason for PACT not having perfect TS? Manual review would be nice to show if the failures are due to z3 timeouts or LLMs synthesizing the ADTs incorrectly.
* What is the average number of CVTs generated per test per dataset?
* Please report temperature/top-p/max-tokens etc

---

### Official Review · Reviewer_2UTi · 2025-11-01

**Soundness:** 3
**Presentation:** 4
**Contribution:** 3
**Rating:** 4
**Confidence:** 4

**Summary:**

In this paper the authors propose PACT, a testing mechanism to test the ability of LLMs to generate code that adheres to implicit and explicit contracts specified in the prompt. The core idea is that while standard code gen benchmarks like MBPP and HumanEval check for functional correctness, they do not check whether generated code appropriately handles illegal inputs. PACT extracts contracts from the prompted task and generates contract violation tests which it runs with the generated code to ensure that the tests trigger the appropriate assertions in the code. They also use the extracted contracts within the prompt for code gen to generate code that enforces the contracts.

**Strengths:**

1. I like the idea of measuring the ability of generated code to handle unexpected inputs (where I guess expected inputs are denoted via contracts). This is a major gap in existing code validation metrics, and makes sense to include in wider code generation evaluations.
2. The use of SMT to generate CVTs that trigger combinations of contract violations is a nice and elegant way of ensuring the thoroughness of the test cases generated by the PACT.
3. The evaluation is thorough as far as the choice of models is concerned, though I do have some other issues with the evaluations presented in the paper (detailed in weaknesses).

**Weaknesses:**

1. I feel like the evaluation does not explore the full effect of PACT on code generation. The most obvious is: how does contract aware generation impact the functional accuracy of the generated code? The tables do not show numbers for pass@1 for vanilla code generation anywhere, so it is hard to factor the overall effect of PACT on whether the final code generated is correct or not. It may very well be the case that adding the contract to the prompt pollutes it enough that the code generated may be less functionally correct even if it satisfies the contracts. I would like to see empirically that adding contract aware generation does not negatively impact the pass@1 metric, or if it does, a thorough discussion on why that happens.

Another issue I see is that there is no discussion on how well the contracts align with the actual task. As per my understanding, neither AVC not TS actually measure whether contracts were correctly extracted; rather they measure whether the extracted contracts are sufficiently captured by the test cases. Measuring alignment is indeed hard, but either a discussion, or some effort to that end, should have been included (e.g. maybe use an LLM-as-a-judge to see if contracts align with the prompt).

2. I would have also liked to see an analysis of the complexity of extracted contracts and the effects of ambiguity in the prompt. For example, in Fig 1, the prompt just wants to detect duplicate values in a sequence, but the extracted contract enforces that each number is an integer, whereas floats should ideally be allowed as well. Actually, the prompt only specifies that it wants duplicate values, so is it just the use of the variable 'nums' in the prompt that triggers the contract that every element needs to be an integer? This seems overly strict. Maybe there should be some analysis of how often the extracted contracts over-specify or under-specify the actual requirements from the task.

**Questions:**

See weaknesses.

Also some suggestions:
1. I see appendix C that partly addresses my first weakness, but it is not enough, this needs to be a proper and thorough evaluation (and it is not that hard).
2. Are there classes of code generation problems where PACT will not be useful? How effective do you think it will be in generating code files with modules, functions, classes, etc.? What would contracts look like in those cases?
3. Are there cases where the test cases generated by PACT contradict the ground truths of HumanEval or MBPP?

---

### Note · Authors · 2025-11-21

I have read and agree with the venue's withdrawal policy on behalf of myself and my co-authors.